# Extracellular Matrix Deposition Defines the Duration of Cell Sheet Assembly from Human Adipose-Derived MSC

**DOI:** 10.3390/ijms242317050

**Published:** 2023-12-01

**Authors:** Valentina S. Glazieva, Natalya A. Alexandrushkina, Peter P. Nimiritsky, Maria A. Kulebyakina, Roman Yu. Eremichev, Pavel I. Makarevich

**Affiliations:** 1Institute for Regenerative Medicine, Medical Research and Education Center, Lomonosov Moscow State University, 27-10, Lomonosovskiy av., Moscow 119192, Russia; glazievavalentina@mail.ru (V.S.G.); n.alexandrushkina@gmail.com (N.A.A.); nimiritsky@gmail.com (P.P.N.); eremichevry@my.msu.ru (R.Y.E.); 2Faculty of Medicine, Lomonosov Moscow State University, 27-1, Lomonosovskiy av., Moscow 119192, Russia; coolebyakina@gmail.com

**Keywords:** cell sheet, tissue engineering, mesenchymal stromal cells, extracellular matrix, fibroblast activating protein, myofibroblast

## Abstract

Cell sheet (CS) engineering using mesenchymal stromal cells (MSC) draws significant interest for regenerative medicine and this approach translates to clinical use for numerous indications. However, little is known of factors that define the timing of CS assembly from primary cultures. This aspect is important for planning CS delivery in autologous and allogeneic modes of use. We used a comparative in vitro approach with primary donors’ (*n* = 14) adipose-derived MSCs and evaluated the impact of healthy subject’s sex, MSC culture features (population doubling time and lag-phase), and extracellular matrix (ECM) composition along with factors related to connective tissue formations (α-SMA and FAP-α) on CS assembly duration. Using qualitative and quantitative analysis methods, we found that, in seeded MSCs, high contents of collagen I and collagen IV had a direct correlation with longer CS assembly duration. We found that short lag-phase cultures faster turned to a ready-to-use CS, while age, sex, fibronectin, laminin, α-SMA, and FAP-α failed to provide a significant correlation with the timing of assembly. In detachable CSs, FAP-α was negatively correlated with the duration of assembly, suggesting that its concentration rose over time and contributed to MSC activation, transitioning to α-SMA-positive myofibroblasts and ECM turnover. Preliminary data on cell density and collagen I deposition suggested that the TGF-β1 signaling axis is of pivotal importance for ECM composition and construct maturation.

## 1. Introduction

Multiple cell therapy clinical trials were hindered by the rapid onset of cell death due to anoikis during injection [1] and immediately after transplantation into damaged tissue [2]. To overcome this challenge, cell sheet (CS) technology was developed by T. Okano as a delivery method using tissue engineering devoid of synthetic scaffolds or animal-based carrier materials [3]. CSs are minimal tissue-like constructs comprising viable adherent cells bound by an extracellular matrix (ECM) produced by them over time in vitro. Besides ECM, ready-to-use CSs also bear soluble growth factors, cytokines, and extracellular vesicles trapped within matrix meshes [4].

After transplantation within CSs, cells preserve their activity and are retained at the site of application for a significant period of time, ranging from days to weeks. Contrasting to widely used spheroids, cells in CSs have access to relatively evenly distributed nutrients and oxygen, providing homogeneous conditions for all cellular elements [5,6].

CSs from mesenchymal stem/stromal cells (MSC) of different tissue origins have gained significant attention as a method for tissue engineering in regenerative medicine. The multipotency of MSCs and rich ECM composition are properties that make CSs from MSCs a valuable utility for treating connective tissue defects (e.g., bone fractures and cartilage degeneration), mucosal ulcers, skin lesions, and cardiac disease [7]. The prospects of CSs from MSCs are beyond doubt, yet certain practical challenges may limit the widespread use of this technology. 

One critical point is related to the standardization of CS assembly that may take from 3 to 15 days to be obtained from harvested primary MSCs. The heterogeneity of MSC features, as well as donors’ features (age, sex, and comorbidity), may affect the MSCs’ properties and determine efficacy of CS assembly. Furthermore, human tissue material (bone marrow and subcutaneous or visceral fat) yields a heterogeneous pool of primary cells, even when commercial reagents or GMP protocols of isolation and culture are used [8]. Eventually, one may fail to obtain a ready-to-use CS within a specific time frame—e.g., by the day of planned surgery or as soon as possible in burn or emergency trauma patients. We failed to find published studies providing insights into feasibly evaluated determinants that impact CS assembly duration, indicating a gap of knowledge that may be of importance for the practical application of this approach.

## 2. Results

### 2.1. Lag-Phase Duration in Seeded MSC Culture Directly Correlates with CS Assembly Duration

During CS assembly from dense MSCs, monolayer cells proliferate and accumulate an extracellular matrix (ECM), which eventually results in spontaneous detachment once the contractile forces within the CSs overcome adhesion. Ready-to-use CS was a visible construct available for detachment from the cultural dish surface without a loss of integrity (Figure 1A). All evaluated MSC cultures and resulting CSs were morphologically comparable (Figure 1B). An analysis of the cultural characteristics demonstrated that the CS assembly duration, surprisingly, did not correlate with the proliferation rate of the seeded MSCs assessed as the population doubling time (PDT) (Figure 1C). However, we found a direct correlation of CS assembly duration and seeded MSC lag-phase in the early period of assembly. Lag-phase is a period required for MSCs to adhere, flatten, and produce ECM to create an environment for proliferation and growth (Figure 1D). Basically, MSC cultures with a short lag-phase showed a faster assembly of CSs ready for detachment. 

### 2.2. MSC Donor Sex Fails to Demonstrate Impact on CS Assembly Duration 

We suggested that donor sex might influence the MSC proliferation and secretory function in vitro in healthy subjects and found that in donor population (*n* = 14) male-to-female distribution was equal to 54:46%, respectively, and found no evidence for sex impact on CS assembly duration (Figure 2).

### 2.3. Collagen I, but Not Fibronectin Contents in Seeded MSC Directly Correlates with Duration of CS Assembly 

Matrix deposition rate is considered to be a significant factor contributing to CS assembly and we performed immunofluorescent visualization of collagen I and fibronectin as its major components. We used a dense monolayer formed by day 2–4 after seeding and attached CSs for two histological procedures—with or without permeabilization with Triton X-100. Under non-permeabilized conditions, proteins deposited in the extracellular compartment are visualized, while permeabilization allows antibody binding with both intra- and extracellular ECM. Thus, permeabilized samples were termed “total ECM”, suggesting that accumulated yet not secreted proteins were visualized as well.

Monolayer MSCs demonstrated minimal extracellular deposition of collagen I, while in assembled CSs, it was detected in the extracellular part (non-permeabilized construct), demonstrating a dramatic accumulation of ECM over time (Figure 3A, “Extracellular ECM”). Using permeabilization, we showed a high intracellular content of collagen I in both monolayer MSCs and CSs, indicating the ongoing production of ECM, even after assembly (Figure 3A, “Total ECM”).

Strand-shaped extracellular deposits of fibronectin were detected in MSC monolayers and assembled CSs (Figure 3A, “Extracellular ECM”). In non-permeabilized monolayers, fibronectin was visualized as a network of thin extracellular filaments (Figure 3A, “Extracellular ECM”), while in CSs, the strength of the signal was vividly higher, covering the field of view. This might also have resulted from the CS multilayered structure we have previously shown using confocal microscopy [9,10,11]. The use of permeabilization had minimal impact on the visualized patterns in the monolayers and CSs (Figure 3A, “Total ECM”), suggesting an even distribution of fibronectin between the intra- and extracellular compartments.

For a semi-quantitative analysis, we used dot-ELISA assay for collagen I and fibronectin in lysates of MSC monolayers and CSs. We found a significant direct correlation (r = 0.657; *p* = 0.014) of collagen I contents in monolayers and the eventual duration of CS assembly from MSCs of a specific donor (Figure 3B, left plot). Thus, MSCs with higher contents of collagen I in their monolayers showed a longer CS assembly period. An analysis of the fibronectin in MSCs showed a non-significant trend (*r* = −0.399; *p* = 0.157) for an inverse correlation between fibronectin contents and CS assembly duration, suggesting that MSC cultures with initial high fibronectin production tend to form CSs faster (Figure 3B, right plot).

### 2.4. Contents of Basement Membrane Component Collagen IV, but Not Laminin in Seeded MSC Directly Correlates with CS Assembly Duration

Laminin and collagen IV are typically attributed to the basement membrane underlying epithelial cells, yet their deposition has a pivotal role in mechanical properties, including ECM viscosity and the formation of innate tensile forces in tissues [12,13]. The immunofluorescent labeling of laminin using the same approach as described above demonstrated that its extracellular deposition in both MSC monolayers and CSs was barely detectable (Figure 4A, “Extracellular ECM”). Sporadic laminin-positive cells with a cytoplasmatic signal were attributed to the weak permeabilizing impact of the 4% paraformaldehyde used for the culture fixation. In contrast, after permeabilization, we found a high intracellular accumulation of laminin in both cultures (Figure 4A, “Total ECM”). Thus, laminin was predominantly accumulated inside MSCs, which was concordant with the lack of correlation of its relative amount and CS assembly duration (*r* = 0.414; *p* = 0.159) found using dot-ELISA (Figure 4B).

The collagen IV signal in the extracellular compartment of monolayer MSCs showed a low intensity with an increase in detachable CSs (Figure 5A, “Extracellular ECM”). After permeabilization, we observed a visible accumulation of collagen IV inside the cells in both monolayers and CSs, which suggested a peculiarly common profile of deposition for two collagens—type IV and type I—in monolayer culture characterized by intracellular accumulation and a drastic increase in its amount once CSs are formed (“Total ECM” in Figure 5A and Figure 3A, respectively). The dot-ELISA results demonstrated a direct correlation (*r* = 0.72; *p* = 0.007) of collagen IV relative contents with CS assembly duration, indicating that MSCs with a high production of this ECM component took longer to form a CS (Figure 5B).

### 2.5. Factors Related to Connective Tissue Do Not Correlate with Assembly Duration Yet May Play a Role in Assembled CS Structure Support

#### 2.5.1. Prevalence of α-SMA-Positive Cells and α-SMA Contents Fail to Correlate with CS Duration Assembly

Among cells that actively produce ECM and contribute to connective tissue formation, myofibroblasts are of pivotal importance. Furthermore, MSCs may undergo differentiation to myofibroblasts under certain conditions, including spontaneous conversion in vitro [11]. We evaluated α-smooth muscle actin (α-SMA) as an intracellular marker of the myofibroblast phenotype in monolayer MSCs and assembled CSs. We found a drastic increase in α-SMA-positive cells in CSs, which may contribute to their contraction used as visual endpoint. This was supported by manual counts of α-SMA-positive cells using DAPI-stained nuclei as an additional cue (Figure 6A). However, an analysis of CS assembly duration and α-SMA contents measured with dot-ELISA (Figure 6B) or α-SMA-positive cell counts (Figure 6C) failed to detect significant correlations. Thus, despite an obvious increase in α-SMA-positive cells, we found no evidence for their potential role in defining the time required for CS maturation.

#### 2.5.2. FAP-α in Seeded MSCs Fails to Correlate with CS Assembly Duration Yet Seems to Be Involved in Their Maturation at Later Terms

Fibroblast-activating protein-α (FAP-α) is a surface serine protease expressed in developing and healing tissues, playing a crucial role in the regulation of stromal cells’ activation and ECM synthesis/turnover [14]. We failed to find a correlation (*r* = −0.409; *p* = 0.362) between the FAP-α contents in monolayer MSCs and CS assembly duration (Figure 7D, “Monolayer MSC”). We also assayed FAP-α in assembled CSs as far as this protease can modulate ECM integrity and found a strong inverse relationship (*r* = −0.92; *p* = 0.003) between FAP-α contents and duration of assembly (Figure 7D, “Cell sheet”). Despite failing to be a predictor of assembly duration, FAP-α increase in ready-to-use CSs hinted its role in ECM maturation and possible support of construct integrity.

### 2.6. Inhibition of TGF-β Signaling Disrupts CS Maturation and Reduces Collagen I Deposition within CSs and Does Not Impact α-SMA Contents

To understand the potential mechanism behind CS maturation and ECM accumulation, we used A-8301 (Sigma-Aldrich, USA)—a potent inhibitor of TGF-β type I receptor at 1 μM. We added it at different time-points to provide the inhibitory impact of short- or long-term duration and, on day 14, we found a time-dependent impairment in cellular mass accumulation (Figure 7, “Hematoxylin”) and collagen I deposition in the extracellular compartment of the CSs (Figure 7, “Collagen I”). 

## 3. Discussion

A comparative analysis using a parallel cultures design (Appendix A) allowed to identify the parameters that may determine the timing required for CS assembly.

Previously, we demonstrated the activation of MSC proliferation at later terms of CS assembly (days 9–12) [9]. The relation between MSC proliferation and rapid CS formation seemed obvious, but our data revealed no significant correlation between the CS assembly duration and PDT (Figure 1C). We were limited to PDT assessment during the early-stage formation of dense monolayers, which may be partially relevant to later events. The transition of monolayers to CSs may impact the cell cycle, leading to the synchronization of the culture and modulation of division rate. Unfortunately, little is known about the cell cycle changes in CSs, besides sporadic studies like one by Hirata et al., who found a Rac1-dependent drop in proliferation and motility in cell sheets from keratinocytes [15]. This gap of knowledge in regard to MSC properties warrants additional studies using cell cycle imaging during CS maturation at days 5–6 after seeding. 

We also found that MSCs with a shorter lag–phase period acquired the properties of a detachable CS much faster (Figure 1D). A substantial lag-phase in MSCs reflected the time needed to adhere and accumulate the solid and soluble components required to initiate mitosis. Our data support an assessment of the lag-phase in seeded MSCs to receive a preliminary evaluation of the expected assembly duration. Despite not being an obvious rate-limiting step, the lag-phase seems a feasible parameter for evaluation during the early stages of MSC culture.

We analyzed the major proteins that comprised the interstitial matrix—collagen I and fibronectin (Figure 3). Collagen I is the most abundant protein in the human body, accounting for 25% of the body, while in the skin, its content exceeds 60% [16]. In addition to type I collagen, high concentrations of fibronectin are also present in the extracellular matrix (ECM) of connective tissues. 

The collagen I and fibronectin contents assayed in the monolayer MSC cultures demonstrated opposite correlations with the duration of CS assembly (Figure 3). However, only a direct correlation for collagen I and assembly duration reached statistical significance (Figure 3). Our data in permeabilized monolayer cultures (Figure 3A) indicated that collagen I was deposited mainly inside MSCs, while fibronectin was secreted and detected in the extracellular compartment. This partially explained the results of the correlative semi-quantitative analysis. Indeed, it relied on dot-ELISA in lysates, so it failed to discern between intra- and extracellular compartments, detecting the total collagen I in the samples. However, this does not diminish the importance of using collagen I contents in monolayer MSCs as a predictive marker of long- or short-term CS assembly.

The stiffness of the ECM enriched by collagen I and fibronectin is associated with the transition of stromal cells to myofibroblasts [17]. This phenomenon is pivotal for in vivo wound healing due to the vivid contractile ability of myofibroblasts mediated by their cytoskeleton. A landmark feature of the transition to myofibroblasts is an increase in α-smooth muscle actin (α-SMA), which precedes the activation of reparative fibrosis [18]. In our study, the ECM-mediated activation of the transition to myofibroblasts might have played a role in CS contraction, which we used as an objective event indicating the completion of assembly. However, the evaluation of α-SMA contents and prevalence of α-SMA-positive cells (Figure 6A–C) failed to reveal significant correlations between both parameters and CS assembly duration. 

We assessed collagen IV and laminin as important components of the basement membrane—a specialized ECM structure delimiting connective tissues from other cell types providing adhesion, mechanical stability, and elasticity. In addition to its structural function, the basement membrane provides a filter, plays a barrier role for large molecules, and controls the cellular phenotype [19]. 

The analysis of laminin revealed no correlation between its content and CS assembly duration (Figure 4). This is explained by its intracellular deposition in both monolayers and CSs with a minor presence in the extracellular space. Thus, its contribution to the formation of CSs was of minor value, while it may influence other processes—e.g., the organization of the basement membrane upon the contact of MSCs with endothelial cells, which has been observed in many co-culture studies [20]. Interestingly, a direct correlation of collagen IV and duration of assembly (Figure 5) was similar to collagen I, despite its different roles: while collagen I is an ubiquitous structural protein, collagen IV is confined exclusively to the basement membrane [21].

Generally, our findings on laminin and collagen IV indicate that their deposition within CSs plays a minor role in their formation, as far as both were deposited intracellularly and were barely detectable without permeabilization (Figure 4 and Figure 5). Nevertheless, their deposition may contribute to increase in MSC number, as far as collagen IV enhances the FAK-ERK signaling pathway associated with the activation of MSC proliferation [22].

We were the first to assay fibroblast activation protein alpha (FAP-α) in mature CSs from human MSCs. An evaluation of FAP-α was conceived to explain the MSC transition to myofibroblasts and the contraction that occurs during CS maturation. 

However, its contents in monolayer MSCs did not correlate with the duration of CS assembly (Figure 6D). However, we found that FAP-α was higher in CSs with rapid assembly (Figure 6D), which hinted that increase in FAP-α occurred at later time points. This assumption was concordant with an α-SMA increase in CSs compared to monolayers due to the known role of FAP-α in stromal to myofibroblast transition. The mechanism of this regulatory circuit remains enigmatic, yet efforts are being made to establish molecular pathways that connect ECM signaling and FAP-α expression in human MSCs [23]. The peptidase activity of FAP-α may also contribute to deposition and turnover of mature ECM proteins in CSs [24]. 

Our findings on α-SMA and collagen I suggested that the TGF-β signaling that plays a crucial role in connective tissue morphogenesis and fibrosis after injury [25,26] might be involved in the regulation of CS assembly. Using an inhibitory analysis, we demonstrated that the blocking of TGF-β type I receptor resulted in a drastic drop in cellular mass accumulation (Figure 7, “Hematoxylin”), which obviously resulted in a lack of high-density areas within the CSs. Our previous data suggest such formations as a morphological sign of mature CSs and we showed that MSCs within high-density areas bear a transcriptomic “fingerprint” of a condensed mesenchyme state with a cell fate commitment towards bone and cartilage, but not adipocyte lineage [9]. Thus, TGF-β signaling regulates MSC self-organization, involving ECM deposition, which drastically dropped after inhibitor addition for 12 days (Figure 7).

## 4. Materials and Methods

### 4.1. Human Adipose Tissue MSC Source and Culture

Previously characterized human adipose-derived MSCs isolated from voluntary donors (*n* = 14) were obtained from the National depositary bank of living systems (Lomonosov Moscow State University, Moscow, Russia). All procedures with tissue samples from patients were performed in accordance with the Declaration of Helsinki and approved by the Local Ethical Committee, Medical Research, and Education Center, Lomonosov Moscow State University (IRB00010587), protocol #4 (2018). Samples of human adipose-derived MSCs were collected from the Cryobank of the Institute for Regenerative Medicine of Lomonosov Moscow State University (collection ID MSC_AD_MSU, www.human.depo.msu.ru, accessed on 28 November 2023). All MSC samples were previously characterized for consensus MSC criteria, including morphology and adhesion, immunophenotype, and trilinear differentiation potency at passage 2–3. Vials were thawed and cells were transferred to a tube with DMEM/F12 medium (Gibco^TM^, TermoFisher Scientifc, Waltham, MA, USA) supplemented with 10% fetal bovine serum (FBS; HyClone, Cytiva, Marlborough, MA, USA), 100 U/mL of penicillin, and 100 µg/mL of streptomycin (Gibco, USA). The cells were centrifuged (5 min, 200 g) to remove dimethyl sulfoxide (DMSO); the supernatant was aspirated, the cell pellet was resuspended in complete DMEM/F12 (10% FBS), and seeded at 5 × 10^3^ cells/cm^2^ to culture dishes (Corning, Glendale, AZ, USA). The cells were cultured under standard conditions (37 °C and 5% CO_2_) with a change of medium every 2–3 days and passaged at an 80–90% cell density using 0.05% Trypsin/EDTA solution (Gibco, USA) at a 1:3 ratio. All cells were used for experiments described below at passages 4–5.

### 4.2. Timelapse Monitoring of MSC Cultures

A proliferation analysis was performed using an IncuCyte^®^ Live Cell Analysis System (Essen Bioscience, Ann Arbor, MI, USA). Cells were seeded to 12-well plates (Corning, USA) at 5 × 10^3^ cells/cm^2^ for 96 h. Timelapse images were captured every 4 h to plot the growth curves reflecting the confluency (percentage of area occupied by cells) in each well of a 12-well plate. Using obtained micrographs of each MSC culture, we analyzed cell morphology over time and determined the duration of the lag-phase. More than two mitotic divisions in the field of view (10× objective) indicated the end of the lag-phase and onset of a subsequent exponential growth phase. The population doubling time (PDT) of the MSCs was calculated for each cell culture at the stage of exponential growth by dividing the natural logarithm of 2 by the exponent of growth.

### 4.3. Assembly of Cell Sheets from MSC

For the reader’s convenience, a diagram of the described experiments is presented in Appendix A. The assembly protocol for tissue-engineered constructs from MSCs involves a long-term culture in complete growth medium until there are constructs that can be freely detached from the culture plastic. MSCs were seeded to uncoated culture plastic at 50 × 10^3^ cells/cm^2^ in complete growth medium (DMEM/F12, 10% FBS) supplemented with ascorbic acid (50 μg/mL) and cultured with medium replacement every 2 days for up to 11 days [27]. We suggested the spontaneous detachment of CSs as an objective endpoint of assembly completion. Briefly, using daily observations, we evaluated the CS assembly duration as the number of days after seeding until the spontaneous detachment of CSs. We used parallel monolayer cultures and undetached CS specimens for histology and protein assays. The majority of the monolayer cultures were analyzed at days 2–4 upon reaching confluency. CS assembly duration varied between donors, showing a median duration of 8 days with a minimum of 5 and maximum of 11 days. 

### 4.4. Immunolabeling Procedures

For immunofluorescent labeling, the MSC sheets or monolayers were washed three times with warm PBS and fixed with 4% formaldehyde in PBS for 10 min. For the labeling of intracellular proteins, cultures were permeabilized with 0.2% Triton X-100 for 10 min and washed three times with PBS for 5 min. Blocking was performed with 1% bovine serum albumin (BSA) with the addition of 10% non-immune serum of a secondary antibodies donor. After blocking, the samples were incubated overnight at 4 °C with primary rabbit polyclonal antibodies vs. human collagen I (4 μg/mL), collagen IV (3.5 μg/mL), fibronectin (3.5 μg/mL), α-SMA (0.5 μg/mL), or laminin (2.5 μg/mL) (all from Abcam, Cambrige, UK) in blocking buffer. Afterwards, the samples were washed three times with PBS for 5 min and secondary antibodies conjugated with Alexa Fluor 488 or Alexa Fluor 594 (Invitrogen, Waltham, MA, USA) at 2 μg/mL were added for 1 h at room temperature. Nuclei were labeled by 4′,6-diamidino-2-phenylindole (DAPI, Sigma-Aldrich, St. Louis, MO, USA) for 30 min in the dark and then washed 3 times with PBS for 5 min. Image acquisition was performed on a DMi8 inverted microscope (Leica Microsystems, GmbH, Wetzlar, Germany).

### 4.5. Inhibitory Analysis In Vitro

We used A-8301 (Sigma-Aldrich, USA)—a potent inhibitor of TGF-β type I receptor (ALK5-TD)—at 1 μM added to the MSC culture medium starting from different time-points during CS assembly. After incubation for 2, 4, 7, or 12 days, we used the assembled CSs at day 14 for the hematoxylin staining or immunolabeling of extracellular collagen I. 

### 4.6. Hematoxylin Staining 

The assembled CSs on the culture plates were fixed with 4% formaldehyde (Panreac, Darmstadt, Germany) for 10 min, permeabilized by 70% isopropyl alcohol for 3 min, washed with distilled water, stained with hematoxylin (DAKO, Santa Clara, CA, USA) for 30 s, then gently washed with tap water for 3 min, and analyzed with microscopy.

### 4.7. Estimation of α-SMA-Positive MSC Prevalence

An analysis of the α-SMA-positive MSCs was carried out using the ImageJ program (NIH, Bethesda, MD, USA, version 2.14.0) in corresponding immunolabeled cultures. The total number of nuclei and the number of α-SMA-positive MSCs were counted in the micrographs to obtain their percentage.

### 4.8. Protein Extraction, Dot-ELISA, and Densitometry

The total protein was extracted from the MSC cultures with Laemmli buffer (BioRad, Hercules, CA, USA). The protein contents within the lysates were assayed using the Pierce^TM^ BCA Protein Assay Kit (Invitrogen, USA), according to the manufacturer’s instructions. Normalization by the DNA contents in the lysed samples was performed using PicoGreen DNA dye (Thermo Fisher Scientific, Waltham, MA, USA), as previously described by our group for a similar application to compare MSC sheets and monolayer cultures and account for differences in cell numbers [9]. Due to high-molecular-weight proteins within the ECM of the CSs, we decided to avoid Western blotting and used a semi-quantitative analysis of the lysates using dot-ELISA. Briefly, the lysed samples (1 μL per dot) were pipetted in duplicate to a nitrocellulose membrane and dried at room temperature. The dotted membrane was sliced and corresponding parts were sequentially incubated in a 5% non-fat milk solution in PBS containing primary antibodies vs. human collagen I, collagen IV, laminin, fibronectin, α-SMA, or FAPα (all from Abcam, UK) at 1:1000 dilution followed by a corresponding secondary HRP-conjugated antibody. After washing in Clarity ECL Solution (Bio-Rad, Hercules, CA, USA), a substrate was added to visualize the HRP signal with image acquisition using ChemiDoc Touch (Bio-Rad, USA). For normalization, we used serially diluted lysates of a known number of cells from a reference MSC culture (ASC52telo, ATCC, Manassas, VA, USA) which has a stable and detectable production of ECM components [9], and a parallel duplicate membrane was stained with Amido Black dye (Sigma Aldrich, USA). 

### 4.9. Statistical Analysis 

Data were processed using GraphPad Prizm 8.0.1. and Microsoft Excel software, version 2016. Descriptive statistics included a normality assessment (Shapiro–Wilk test) and calculation of the mean and standard error of the mean (all data are presented as M ± SEM unless otherwise specified). The differences between the groups were evaluated using Student’s *t*-test or Mann–Whitney test for parametric or non-parametric distributions, respectively. For a correlation analysis, the Pearson linear correlation coefficients was calculated. All differences and correlations were deemed significant at *p* < 0.05.

## 5. Conclusions

Our study is the first to investigate the potential factors impacting CS assembly and suggests the use of an early-stage evaluation of ECM composition to assess the time required for the completion of assembly. From our sample size, the most evident correlations were with collagen I and IV production and MSC lag-phase routine measurement. We also stress that the semi-quantitative dot-ELISA method is a feasible and cheap approach once the data are normalized using the total DNA or protein data obtained from assayed samples. 

These data were obtained using a relatively small sample size (*n* = 14), which may impact the study power. Nonetheless, a study in a larger population may reveal additional significant factors based on data trends. Consequently, other correlations should not be conclusively dismissed; thus, further recruitment of adipose tissue donors will provide an opportunity to strengthen the results and identify additional factors. 

We also demonstrated the role of TGF-β in CS structural maturation (at least for collagen I), which may suggest a study of the TGF-β production in MSCs as a future direction to search for a feasible biomarker related to CS assembly.

## Figures and Tables

**Figure 1 ijms-24-17050-f001:**
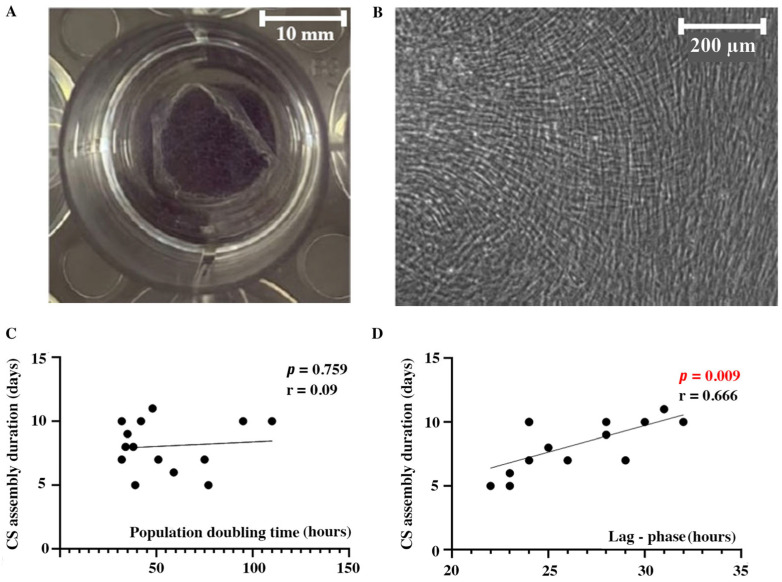
Cell sheets’ morphology and correlation of cultural characteristics with assembly duration. Macroscopic photo of CS after partial detachment from a 12-well plate (**A**) and representative micrograph of CS from human adipose tissue MSC (**B**) demonstrating cell morphology (phase-contrast microscopy). Dotted plots represent Pearson correlations of CS assembly duration with population doubling time (**C**) and lag-phase (**D**) after seeding. Individual data points and trend-line are plotted.

**Figure 2 ijms-24-17050-f002:**
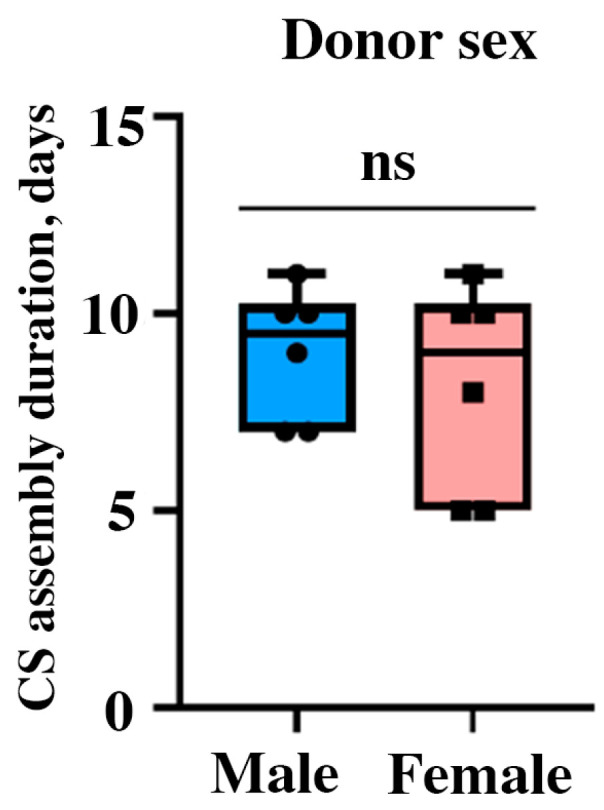
Impact of donor’s sex on CS assembly duration from primary; ns—non-significant, Student’s *t*-test; box limits denote standard deviation, horizontal bar inside the box stands for mean, “whiskers”—for minimal and maximal values; individual data points are denoted by dots or squares (may overlap; *n* = 14).

**Figure 3 ijms-24-17050-f003:**
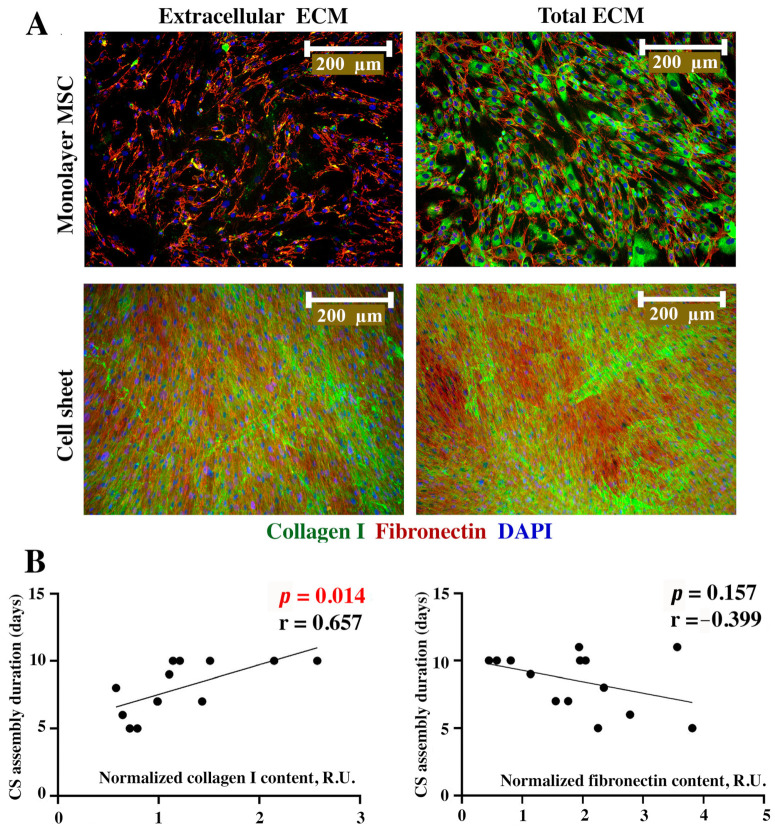
Immunofluorescent labeling of collagen I and fibronectin in monolayer MSCs and CSs (**A**) and corresponding Pearson correlation analysis (**B**) of collagen I or fibronectin content in MSC monolayer and duration of CS assembly. Individual data points and trend-line are plotted. R.U.—relative units.

**Figure 4 ijms-24-17050-f004:**
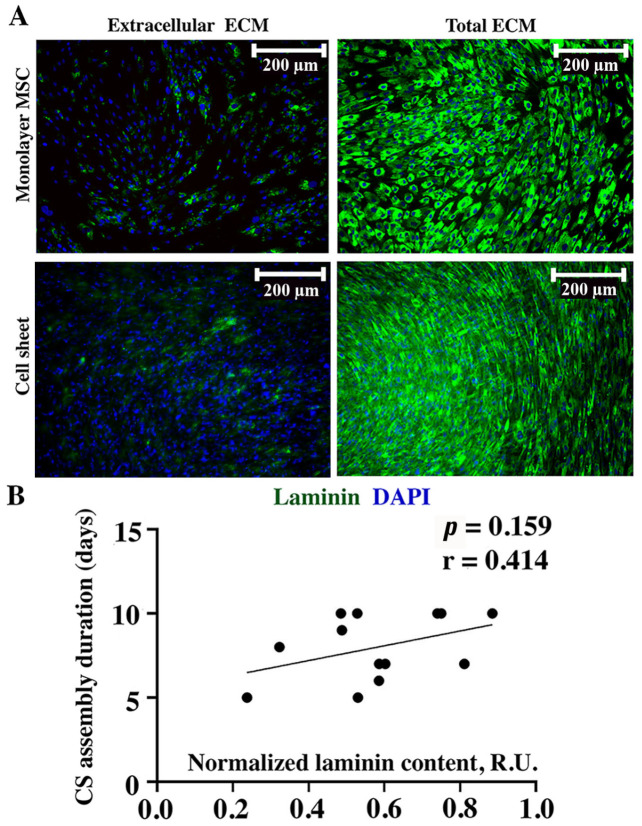
Immunofluorescent labeling of laminin in monolayer MSCs or CSs (**A**) and corresponding Pearson correlation analysis (**B**) of laminin contents in MSC monolayer and duration of CS assembly. Individual data points and trend-line are plotted. R.U.—relative units.

**Figure 5 ijms-24-17050-f005:**
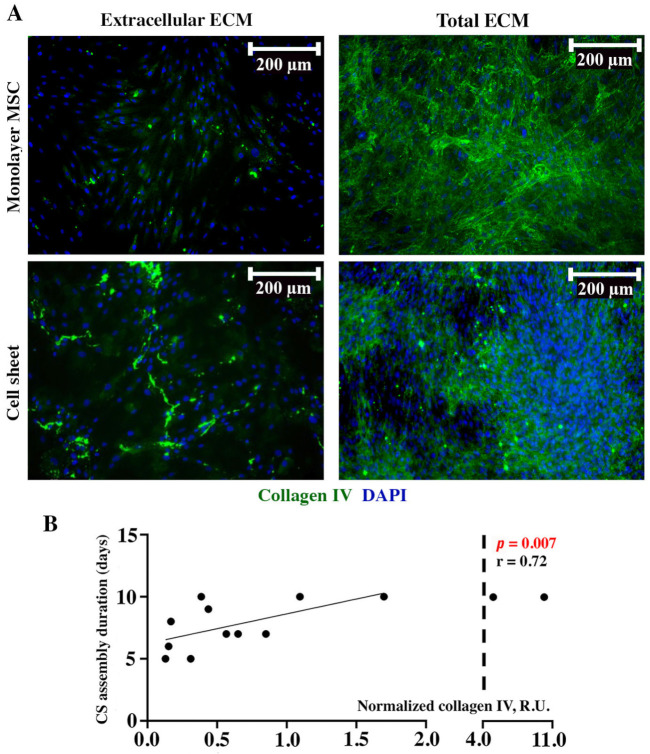
Immunofluorescent labeling of collagen IV in monolayer MSCs or CSs (**A**) and corresponding Pearson correlation analysis of collagen IV contents in MSC monolayer lysate and CS assembly duration (**B**). Individual data points and trend-line are plotted. R.U.—relative units.

**Figure 6 ijms-24-17050-f006:**
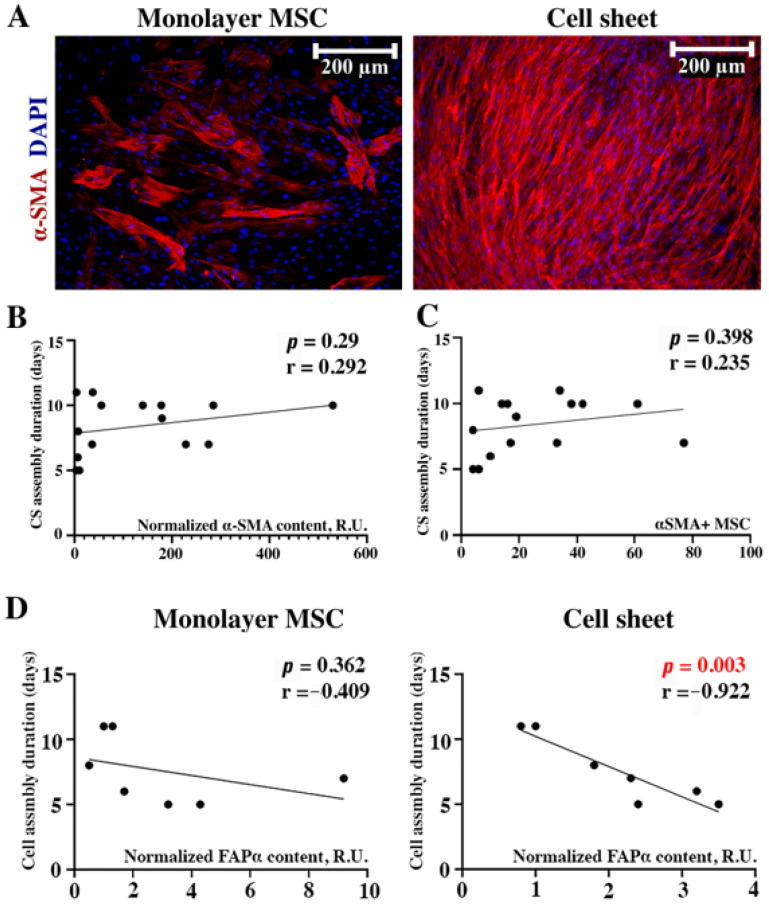
Evaluation of myofibroblast prevalence and FAP-α content in CS assembly duration. Immunofluorescent labeling of α-SMA in monolayer MSCs and CSs (**A**). Correlation of α-SMA contents (**B**) or myofibroblast counts (**C**) in monolayer MSC sand CS assembly duration. Impact of FAP-α was detectable (*n* = 7) as an inverse correlation only in assembled CSs (**D**), while in monolayer MSCs, it failed to reach statistical significance (dotted plots represent Pearson’s correlations). Individual data points and trend-line are plotted. R.U.—relative units.

**Figure 7 ijms-24-17050-f007:**
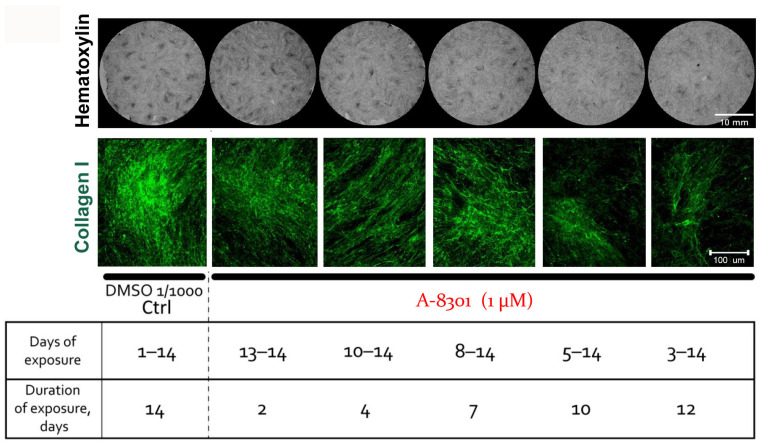
MSC sheet maturation is disrupted by inhibition of TGF-β type I receptor. Both cellular mass accumulation (detected by hematoxylin staining) and collagen I deposition were impaired in a time-dependent manner by exposure to A-8301 (1 μM) for 2, 4, 7, 10, or 12 days during a 14-day period of CS assembly.

## Data Availability

Due to donor confidentiality we do not disclose raw data.

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
