# Peer review of "Extracellular Matrix Deposition Defines the Duration of Cell Sheet Assembly from Human Adipose-Derived MSC"

_ijms, 2023, doi:10.3390/ijms242317050_

Round 1

Reviewer 1 Report

Comments and Suggestions for Authors

To authors

Below are my comments on the various points:

- An important part of the information is mentioned, but not reported, add figure 7.

- The in vitro cell study protocol should be described in detail to allow a better understanding of the study design and results. I believe it is important to report in this article the specific characteristics of the cells used. This information must include cell density, medium composition, and frequency of changes in the medium, but should also include information on number of cells passages used.

- The paper contains errors and typos (see page 8 line 220) that make it difficult to understand and distort its intended meaning. I encourage authors to reread carefully and correct any grammatical errors.

- The document contains several acronyms, which create confusion and make reading the manuscript difficult. Please define acronyms the first time and use them alternated together with the long name to make the manuscript more fluid and understandable.

Comments on the Quality of English Language

Minor editing of English language required

Author Response

To authors

Below are my comments on the various points:

- An important part of the information is mentioned, but not reported, add figure 7.

Reponse We appreciate your comment and added Figure 7 with additional data we recently obtained on impact of TGF-b disruption on CS maturation and accumulation of Collagen I. We feel that this might be a more valuable addition than dot-ELISA of FAP-a or a-SMA which basically are presented as routinely used dot-blotting with densitometry. Thus, we suggest to deepen on putative mechanism involving a widely known morphogen – TGF-b1. Please, see Section 3.6 and Fig. 7

- The in vitro cell study protocol should be described in detail to allow a better understanding of the study design and results. I believe it is important to report in this article the specific characteristics of the cells used. This information must include cell density, medium composition, and frequency of changes in the medium, but should also include information on number of cells passages used.

Reponse We have provided all requested data in Materials and Methods and thank your for your hint – we forgot to mention the Passage of MSC we used – this was added to the text

- The paper contains errors and typos (see page 8 line 220) that make it difficult to understand and distort its intended meaning. I encourage authors to reread carefully and correct any grammatical errors.

Reponse We appreciate this comment and made numerous typo, style and grammar corrections throughout the text

- The document contains several acronyms, which create confusion and make reading the manuscript difficult. Please define acronyms the first time and use them alternated together with the long name to make the manuscript more fluid and understandable.

Reponse We appreciate this comment and made numerous changes to correct this problem

Reviewer 2 Report

Comments and Suggestions for Authors

The manuscript addresses an important topic in regenerative medicine, exploring the factors that define the timing of CS assembly from primary cultures. The work, particularly the investigation into the relationship of various ECM components and their impact on cell sheets, offers a novel perspective that adds value to the field. However, after a careful evaluation, I have identified specific concerns that, when addressed, could significantly strengthen the paper. Major revisions, especially regarding the age-related information, are recommended.

Specific Comments:

Sample Size: The study relies on a relatively small sample size (n=14), which could potentially limit the robustness of the findings. It might be beneficial to acknowledge this limitation and suggest that some trends might be more pronounced with a larger sample size.

Interpretation of Results: A more nuanced presentation could benefit The statement regarding correlations. A suggested rephrasing is: "From our sample size, the most evident correlations were with collagen I and IV production, etc. Nonetheless, a larger sample might reveal additional significant factors based on data trends. Consequently, other correlations should not be conclusively dismissed.

Age Range Concerns and Subsequent Implications: The age range of your donors is notably narrow, concentrating primarily between 30 and 40 years. This limited span raises concerns about the robustness of age-related findings. Given this, I strongly recommend major revisions to either omit age as a defining factor for this study or clarify its limited scope. Relatedly, paragraph 202-207 does not acknowledge established literature on age's impact on cells. If age-related findings are retained, this section should be more heavily referenced with existing research or be removed.

Figures and Statistical Analysis: It would enhance reader comprehension if you detailed the specific statistical methods used. Furthermore, elaborating on the nature of error bars in the figures will provide additional clarity.

Author Response

The manuscript addresses an important topic in regenerative medicine, exploring the factors that define the timing of CS assembly from primary cultures. The work, particularly the investigation into the relationship of various ECM components and their impact on cell sheets, offers a novel perspective that adds value to the field. However, after a careful evaluation, I have identified specific concerns that, when addressed, could significantly strengthen the paper. Major revisions, especially regarding the age-related information, are recommended.

Specific Comments:

Sample Size: The study relies on a relatively small sample size (n=14), which could potentially limit the robustness of the findings. It might be beneficial to acknowledge this limitation and suggest that some trends might be more pronounced with a larger sample size. Interpretation of Results: A more nuanced presentation could benefit The statement regarding correlations. A suggested rephrasing is: "From our sample size, the most evident correlations were with collagen I and IV production, etc. Nonetheless, a larger sample might reveal additional significant factors based on data trends. Consequently, other correlations should not be conclusively dismissed.

Response: We agree that the sample of 14 donors is quite small. This work is part of a study to search for determinants that influence the assembly of tissue-engineered constructs - given that the literature data on assessing potential factors is very scarce, we tried to evaluate as many parameters as possible and select the most promising ones at the initial stages. Of course, we have made changes to the discussion text to acknowledge the possible limitations of the results presented.

Age Range Concerns and Subsequent Implications: The age range of your donors is notably narrow, concentrating primarily between 30 and 40 years. This limited span raises concerns about the robustness of age-related findings. Given this, I strongly recommend major revisions to either omit age as a defining factor for this study or clarify its limited scope. Relatedly, paragraph 202-207 does not acknowledge established literature on age's impact on cells. If age-related findings are retained, this section should be more heavily referenced with existing research or be removed.

Response: we agree that our sample size is insufficient to state that correlation is non-existent so we altered Discussion section to stress that this piece of evidence has limited power and referenced certain aspects of this problem (see lines 336-348) and also added a note on limitations in Conclusion section.

Figures and Statistical Analysis: It would enhance reader comprehension if you detailed the specific statistical methods used. Furthermore, elaborating on the nature of error bars in the figures will provide additional clarity.

Response: we agree on this comment and have made numerous clarifications in Figure legends and in Materials and methods section

Reviewer 3 Report

Comments and Suggestions for Authors

In this manuscript, the authors conducted a detailed study on the assembly of cell sheet (CS) from mesenchymal stem cells (MSC) for tissue repair and regeneration. The study investigates the impact of various factors, such as ECM composition, MSC heterogeneity, and donor features, on the efficacy of CS assembly. The results are presented in a clear and concise manner, and the authors provide a thorough discussion of the implications of their findings. To make the manuscript contents clearer to the readers, I have the following questions and comments for the consideration of authors:

1.       Please fix the scale number in figures (µ to µm).

2.       One scale bar is missing in an image of Figure 4.

3.       In line 242, is ‘not’ a typo?

4.       In section 4.4, please report the actual concentration of antibodies used for IHC, not dilutions.

5.       How did the author ensure the stemness of the MSCs? Would it be necessary to check the characteristic markers of MSCs, e.g. CD105, CD73 and CD90?

6.       According to the manuscript, the sample size of 14 donors is quite small. Can the authors comment on this and predict if the results would remain the same if the sample size if larger?

Author Response

  1. Please fix the scale number in figures (µ to µm).

Reponse We appreciate this comment and fixed the problem.

  1. One scale bar is missing in an image of Figure 4.

Reponse We appreciate this comment and fixed the problem.

  1. In line 242, is ‘not’ a typo?

Reponse We apologise for a puzzled style and tried to clarify the sentence. We also made numerous changes in the text

  1. In section 4.4, please report the actual concentration of antibodies used for IHC, not dilutions.

Reponse We appreciate this comment and fixed the problem.

  1. How did the author ensure the stemness of the MSCs? Would it be necessary to check the characteristic markers of MSCs, e.g. CD105, CD73 and CD90?

Reponse The MSC cultures used in the work were obtained from the National Depository Bank of Living Systems (Lomonosov Moscow State University, Russia), the cryobank of the Institute of Regenerative Medicine, Moscow State University named after M.V. Lomonosov (collection identifier MSC_AD_MSU, www.human.depo.msu.ru). The procedure for banking MSC lines involves checking the phenotype of isolated cells (positive markers CD73, CD90, CD105, negative markers CD34, CD45, CD14, CD19, and anti-HLA-DR) and their ability to differentiate in canonical directions before freezing. In addition, in other studies we have repeatedly assessed the identity of MSCs after thawing; therefore, in this work, it was decided not to perform additional immunophenotyping of MSCs. However, in Materials and Methods we have added a line about this (please, see lines 72-74)

  1. According to the manuscript, the sample size of 14 donors is quite small. Can the authors comment on this and predict if the results would remain the same if the sample size if larger?

Reponse The work we present is aimed at identifying determinants that can influence the duration of assembly of cell sheets from MSCs, which is important for the practical use of cell products. The search part at the first stages of such studies includes the assessment of many parameters, which significantly increases the number of experimental points, the time it takes to complete experiments, and the volume of required cellular material. In this case, we used sufficient sampling (14 MSC lines) to exclude cell characteristics that did not show correlation. Further recruitment of donors will allow us to concentrate only on those parameters that have yielded promising results. We hypothesize that a larger sample will provide an opportunity to strengthen existing findings and perhaps find additional determinants based on these findings (such as the FAPa finding).

However, we acknowledge that such a small sample could potentially limit the reliability of the results. The text has been amended to state limitations and discuss prospects as the number of donors increases.

Round 2

Reviewer 2 Report

Comments and Suggestions for Authors

I have reviewed your revised manuscript and thank you for your response to my previous comments. However, tt is my firm position that the section of the paper discussing donor age lacks the necessary scientific validity due to the narrow age range of the sample and should be removed. The current claims may not only mislead but potentially impede scientific progress if misinterpreted. Thus, these claims must be excluded from the paper to maintain the integrity of the research.

The remainder of the paper is of high quality and should be published. However, it is imperative that the publication proceeds without the age-related claims to ensure the paper’s contributions are both robust and credible.

Author Response

Dear colleague!
We appreciate your firm stance and dropped the claim indicating that additional work is to be conducted to ensure adequate data and valid conclusion on impact of donor age.

We re-submit the manuscript with corrections made in Review mode leaving the result in the text yet clearly indicating its preliminary nature in Discussion. 

Best regards, Authors

Round 3

Reviewer 2 Report

Comments and Suggestions for Authors

The authors have not adequately addressed my concerns for the paper and are still including the problematic data. My objection is not an issue with statistical power; it is an issue with the participant age range, leading to a conclusion that can not be supported or trusted. 

Once more, I ask that the authors remove all data, suggestions of a trend, and mention of age from this study. This requires the modification of the title, abstract, results and discussion. 

I want to reiterate that the remainder of this paper should be published as is, once this issue has been addressed.

Author Response

Dear colleague!
We apologise for misunderstanding and have removed all data as requested to avoid confusion.

Best regards, Authors 

Round 4

Reviewer 2 Report

Comments and Suggestions for Authors

I thank the authors for making the suggested changes and would recommend publication.